# Rice Origin Tracing Technology Based on Fluorescence Spectroscopy and Stoichiometry

**DOI:** 10.3390/s24102994

**Published:** 2024-05-09

**Authors:** Changming Li, Yong Tan, Chunyu Liu, Wenjing Guo

**Affiliations:** 1School of Physics, Changchun University of Science and Technology, Changchun 130022, China; changming_li0034@163.com (C.L.); liucy@cust.edu.cn (C.L.); 15849623028@163.com (W.G.); 2Engineering Technology R&D Center Changchun Guanghua University, Changchun 130033, China

**Keywords:** fluorescence spectroscopy, rice, origin identification, successive projections algorithm, support vector machine, spectral analysis

## Abstract

The origin of agricultural products is crucial to their quality and safety. This study explored the differences in chemical composition and structure of rice from different origins using fluorescence detection technology. These differences are mainly affected by climate, environment, geology and other factors. By identifying the fluorescence characteristic absorption peaks of the same rice seed varieties from different origins, and comparing them with known or standard samples, this study aims to authenticate rice, protect brands, and achieve traceability. The study selected the same variety of rice seed planted in different regions of Jilin Province in the same year as samples. Fluorescence spectroscopy was used to collect spectral data, which was preprocessed by normalization, smoothing, and wavelet transformation to remove noise, scattering, and burrs. The processed spectral data was used as input for the long short-term memory (LSTM) model. The study focused on the processing and analysis of rice spectra based on NZ-WT-processed data. To simplify the model, uninformative variable elimination (UVE) and successive projections algorithm (SPA) were used to screen the best wavelengths. These wavelengths were used as input for the support vector machine (SVM) prediction model to achieve efficient and accurate predictions. Within the fluorescence spectral range of 475–525 nm and 665–690 nm, absorption peaks of nicotinamide adenine dinucleotide (NADPH), riboflavin (B2), starch, and protein were observed. The origin tracing prediction model established using SVM exhibited stable performance with a classification accuracy of up to 99.5%.The experiment demonstrated that fluorescence spectroscopy technology has high discrimination accuracy in tracing the origin of rice, providing a new method for rapid identification of rice origin.

## 1. Introduction

Rice, as one of the globally important food crops, has great significance for agricultural production and market regulation in terms of its quality and origin identification. Traditional rice identification methods mostly rely on morphological characteristics and chemical composition analysis, but these methods often have problems such as time-consuming and low accuracy.

Scholars at home and abroad have conducted extensive research on the traceability of rice origin. For example, Min Sha et al. used Raman spectroscopy to collect spectral information of rice, combined with classification methods such as PCA and SVM, to accurately identify the origin of rice [1]. Fangming Tian et al. selected representative rice from different regions as experimental samples, used Raman spectroscopy information, and employed methods such as smoothing and differentiation to preprocess the original spectra. They also used PCA and SPA to extract the optimal principal components and effective wavelengths, established a classification model, and achieved good classification results [2]. In addition, spectroscopy technology has also been applied to the traceability of the origin of other agricultural products. Yang Jian et al. identified different rice varieties using laser-induced fluorescence technology combined with PCA and SVM, achieving a classification accuracy of 1.36% [3]. Wu Jingzhu collected hyperspectral images of rice from different origins and varieties, extracted the average spectrum of the region of interest of the single rice grain contour, used principal component analysis to extract feature wavelengths, and established a rapid discrimination model for rice origin based on the AlexNet convolutional neural network [4].

Near-infrared spectroscopy plays an important role in the quality detection of agricultural products. By collecting and analyzing the near-infrared spectra of agricultural products, internal component and structural information can be obtained, and their quality can be evaluated. However, Ma Benxue pointed out that most of the current research results are still in the laboratory stage and still have some distance from practical applications [5]. Hyperspectral imaging technology combines spectroscopy with imaging technology, enabling the simultaneous acquisition of spatial and spectral information of samples. Sun Jun et al. used hyperspectral image deep features to detect the viability level of rice seeds. Through wavelet threshold denoising preprocessing, principal component analysis was used to extract feature variables, and the SVM model was optimized using the gray wolf optimization algorithm. Finally, a classification accuracy of 98.75% was achieved [6]. Mahsa Edris used a hyperspectral imaging system (HSI) to collect original spectral data from three rice varieties in Iran. Pretreatment methods such as detrending (DT), multiplicative scatter correction (MSC), and standard normal variate (SNV) were used. Principal component analysis (PCA) was used to reduce the dimensionality of the samples, and a rice variety classification model was established based on the SOM and k-means clustering algorithms [7].

In terms of fluorescence spectroscopy detection technology, Chen Wei and his colleagues collected fluorescence spectroscopy data of solid powder samples of lily bulbs from four origins: Pingjiang, Zhuzhou, Longshan in Hunan Province and Wanzai in Jiangxi Province. By using the two methods of PCA-LDA and PLS-DA, a discriminant model for the origin of lilies was constructed, with classification accuracies of 93.6% and 95.6%, respectively [8]. This provides an effective traceability method for lily samples from different origins. In terms of traceability of the origin of Baishu, Fang Xin collected three-dimensional fluorescence spectroscopy data of Baishu from Anhui, Hunan and Zhejiang. Combined with the two pattern recognition methods of PLS-DA and KNN, traceability analysis of Baishu samples was carried out. This method can effectively distinguish samples from specific production areas of Baishu, with classification accuracies of up to 80% and 90%, respectively [9].

The working principle of fluorescence spectroscopy: When light of a specific wavelength is irradiated on certain substances, the substances absorb light energy and transition to a higher energy state. As they return to a lower energy state, fluorescence is emitted. By detecting the wavelength and intensity of this fluorescence, information about the structure and properties of the substances can be obtained, thus enabling rapid detection of the quality of agricultural products.

## 2. Materials and Methods

### 2.1. Material Preparation

The rice kernels used in this experiment were provided by Jilin Academy of Agricultural Sciences. Before the experiment, the seeds were stored in a cold storage at a temperature of −18 °C. The variety name of rice is Japonica rice 816. The same variety was planted in eight rice-producing areas of Jilin Province in 2020 (Gongzhuling, Hunchun, Jilin, Meihekou, Qianguo, Songyuan, Taoerhe, Yushu). Samples were selected from intact, ungerminated, and mold-free seeds, with a total of 800 grains. A total of 100 grains were selected from each origin, and then hulled, milled, and sieved with a 100-mesh sieve. The samples are shown in Figure 1. The standard sieving sieve of 100 mesh meets [GB/T 6003.1-2012].

The hulling machine uses an aluminum alloy material rice huller with a hulling rate of more than 99%. The model is JLGJ-45, with a motor voltage of 220 V, a power of 120 W, and a weight of 3.5 kg. It meets the requirements of the new national standards GB 1350-1999 and GB/T 17891-1999. The grinding machine uses the Donlim brand DL-MD18 grinding machine, with a rated voltage of 220 V and a rated power of 150 W, from Dongling Electric Appliances Co., Ltd., Foshan, Guangdong Province, China.

### 2.2. Spectral Equipment and Spectral Acquisition

Hamamatsu FFT-CCD detector is the core component of the research-grade spectrometer Ocean Optics QE65pro. The spectrometer has a detection range covering the spectrum of 200–1000 nm, with an excitation wavelength of 405 nm, a resolution of 0.14–7.7 nm, and a scan count of 5 times. The probe of the spectrometer is located directly above the sample, at a height of 100 mm from the sample, with a field of view of 20°.

The spectrometer is equipped with a USB interface on the side for easy connection to a computer. It also comes with an SMA905 interface and a quartz optical fiber with a light transmission range of up to 200–1100 nm. The core diameter of the optical fiber is 600 μm, and the model is UV600-1.0. In addition, an aluminum alloy optical fiber collimator, model 74UV, is provided. The lens is made of ultraviolet fused silica, with a diameter of 5 mm and a focal length of 10 mm, and is equipped with an SMA905 interface.

The distance between the light source and the center of the sample is 300 mm, and it is angled at 45° from the horizontal plane. Before spectral acquisition, the spectrometer needs to be preheated for 30 min and placed at room temperature for 24 h. The fluorescent spectroscopy detection equipment is shown in Figure 2.

In addition to the QE65pro spectrometer, all the equipment mentioned above is provided by Shanghai Wenyi Photoelectric Technology Co., Ltd, Shanghai, China.

### 2.3. Model Evaluation Criteria

The evaluation parameters of the model mainly focus on two key parameters: the misjudgment rate and the total misjudgment rate. The misjudgment rate, which is the proportion of the number of misjudged samples to the total number of samples, is a key indicator to measure the modeling effect. The lower the misjudgment rate, the better the model performance. As a machine learning algorithm, Support Vector Machine (SVM) performs well in dealing with classification problems, especially when the number of training samples is relatively small. SVM is not sensitive to the dimension of input data, so it can maintain good performance even when the data dimension is high [8]. In the field of food analysis, SVM has been widely proved to be a robust and efficient tool, and has been successfully applied in many studies [8,9,10,11]. In the process of experiment of this article, data preprocessing was performed using Python version 3.9, while model evaluation was implemented on Matlab version 2021a.

### 2.4. Spectral Preprocessing Method

In the process of spectral data collection, due to instrument interference, uneven sample distribution, and the influence of random factors, noise and interference inevitably exist in the original spectral data. To improve the signal-to-noise ratio of the spectrum and reduce the impact of noise, preprocessing of the original spectrum is required. Commonly used spectral preprocessing methods include smoothing filtering (Savitzky–Golay, SG), normalization (NZ), and wavelet transform (WT).

Three preprocessing methods were applied in this paper: NZ, SG and WT. These methods are all based on the Sklearn library of Python. Among them, SG uses a 5-point smoothing algorithm; NZ uses linear normalization technology to linearize the original data and convert it to a specific range, usually [0, 1], with its parameter being MinMaxScaler; WT uses Daubechies-8 wavelet as a basic function.

SG is a commonly used preprocessing method in spectral analysis. It smooths data by fitting a polynomial, which can improve the smoothness of the spectrum and effectively reduce the interference of noise, making the spectral signal clearer and more reliable [12]. NZ is another important preprocessing method. It eliminates the dimensional impact between indicators, enabling data to be compared and analyzed on the same scale. After normalization treatment, spectral data is more concentrated, and the intensity distribution range of the spectrum becomes narrower, which helps to improve the stability and accuracy of the model [13]. WT is a signal processing method with multi-resolution characteristics. It can simultaneously characterize high-frequency unstable signals in the time domain and frequency domain. In spectral data processing, wavelet transform is suitable for smoothing denoising and feature extraction. It can effectively extract useful information from the spectrum and improve the discrimination ability of the model [14].

This article aims to compare and study the processing effects of three different preprocessing methods (SG, NZ, WT) on original spectral data. Through modeling analysis of preprocessed data using linear regression, we hope to find a preprocessing method suitable for rice spectra from different regions to improve the performance and stability of the model.

## 3. Results and Discussion

### 3.1. Spectral Data Preprocessing

#### Selection of Preprocessing Methods

Fluorescence spectroscopy carries a large amount of information, some of which is uncorrelated with modeling and may even interfere with modeling effectiveness. Therefore, it is necessary to screen out the characteristic wavenumbers related to modeling and eliminate useless data. Rice from different regions is generally similar in nutritional composition, but subtle differences in content lead to differences in their intensity. Although the spectral curves of rice from different regions differ in peak intensity, the peak positions are basically the same. Considering the influence of instrument noise on the beginning and end of the spectrum, we choose to retain 1527 bands in the range of 450–800 nm as the original spectrum for subsequent analysis, as shown in Figure 3. Figure 3 shows that there is significant high-frequency noise and peak overlap in the original spectrum, which makes accurate peak analysis difficult. To improve the accuracy of information extraction and discrimination, it is necessary to preprocess the original spectrum.

During the data processing stage, to ensure comparability among samples from different origins and minimize the impact of experimental errors, we applied normalization (NZ) treatment to the spectral data of each sample. This treatment narrowed the intensity distribution range of the spectra while maintaining the shape trend, as shown in Figure 4a. When compared to the spectral curves in Figure 4a, the spectra after NZ-SG treatment exhibited reduced spikes, significantly improved smoothness, and maintained a high degree of similarity to the original spectra, as depicted in Figure 4b. Additionally, NZ-WT treatment effectively removed interference and noise signals from the spectral data, resulting in a more pronounced difference compared to the original spectra. This treatment not only achieved smoothing and denoising of the spectral data, but also successfully extracted key features, as illustrated in Figure 4c.

To discriminate rice originating from different regions, we first collected spectral data for rice samples from various locations using a fluorescence spectrometer. Three spectral preprocessing methods were applied to the fluorescence spectra of rice in this study, and the processed spectral data were used as input variables for a Long Short-Term Memory (LSTM) model. To assess the performance of the model, we typically rely on metrics such as the coefficient of determination (R^2^), mean absolute error (MAE), and root mean square error (RMSE). A higher R^2^ value and lower MAE and RMSE values indicate better preprocessing effects [15]. The specific results of the LSTM correction model are presented in Table 1.

From Table 1, it can be observed that in the training set, the NZ-WT preprocessing method exhibited the highest R^2^ value and the lowest MAE and RMSE values, indicating superior preprocessing performance. Although the R^2^ value for the NZ preprocessing method was slightly lower than that of NZ-WT, the difference was not significant. In contrast, the NZ-SG preprocessing method had the lowest R^2^ value, indicating relatively poorer preprocessing effects. In the test set, the advantages of the NZ-WT preprocessing method were more pronounced, with the highest R^2^ value and the lowest MAE and RMSE values, further validating its superiority in spectral preprocessing.

Analysis suggests that the composition of rice is quite complex, and the content of each component is relatively low, resulting in relatively weak spectral information. While the SG preprocessing can eliminate noise in the data, it may also lead to the loss of spectral features, thereby affecting the classification accuracy of the model. In contrast, the WT preprocessing can effectively suppress noise in the spectrum, improve the signal-to-noise ratio, and highlight useful spectral information. This processing not only improves the utilization of spectral information, but also helps to eliminate the impact of baseline drift on rice spectral data [16]. Therefore, in subsequent studies on rice spectral processing and analysis, spectral data based on NZ-WT preprocessing will be utilized.

### 3.2. Analysis of Fluorescence Spectra of Rice

Significant differences can still be observed in specific wavelength ranges, although rice from eight different origins exhibits high similarity in fluorescence spectra. These differences reveal subtle differences in the structure and content of some chemical components in rice from different origins. After preprocessing and feature extraction, the characteristic information of rice fluorescence spectra was obtained. The main components of rice, such as starch, protein, and fat, exhibit different vibrational spectroscopy information due to differences in their chemical composition, content, and structure. As shown in Figure 5, the fluorescence spectrum of rice is mainly concentrated in the range of 450~800 nm. There are two obvious fluorescence characteristic peaks in the wavelength ranges of 475~525 nm and 665~690 nm, which further verifies that the main nutrients of rice are glutelin, alkaline lignin, zein, and corn starch, as shown in Figure 5. In addition, the results of spectral identification are also consistent with previous studies, confirming the presence of nicotinamide adenine dinucleotide (NADPH) and riboflavin (B2) in rice in the wavelength range of 460 nm~525 nm [17]. These findings not only help us to better understand the chemical composition of rice, but also provide a powerful spectral basis for distinguishing rice from different origins.

### 3.3. Spectral Feature Extraction

Fluorescence spectroscopy contains abundant information, but it also accompanies high data redundancy, which increases the complexity of data processing. In order to reduce experimental errors, remove noise, simplify calculations, and reduce modeling variables, it becomes crucial to perform dimension reduction on spectral data. The purpose of dimension reduction is to represent the original entire spectral data with fewer characteristic variables to minimize information loss.

To achieve this goal, we attempt to adopt wavelength screening methods. This approach can effectively reduce data dimensionality, eliminate useless information, and extract key spectral feature information. Through this method, we can process fluorescence spectroscopy data more efficiently and improve the accuracy and efficiency of analysis.

#### 3.3.1. Extraction of Characteristic Variables Based on SPA

The Successive Projections Algorithm (SPA) is a forward cyclic selection method that determines the optimal sample set by calculating the Root Mean Square Error (RMSE) of multivariate linear regression models for different sample subsets. This algorithm uses the principle of vector projection to select effective wavelengths with minimal redundancy and collinearity.

The SPA uses the Kennard–Stone (KS) method to select 50% of the samples for building the estimation model, with the remaining 25% of the sample data used as a validation set, and the remaining 25% used for validating the accuracy and stability of the prediction model. The model accuracy is comprehensively evaluated by using the model evaluation parameter, root mean square error (RMSE).

When applying SPA to select feature variables, we set the range of the number of feature variables to be selected from 1 to 550. The selection process is based on RMSE. As shown in Figure 6, as the number of variables increases, the RMSE value first experiences a rapid decline, followed by a gradual slowdown in the rate of decline. This indicates that in the initial stage, increasing the number of variables can significantly improve the prediction performance of the model; however, when the number of variables increases to a certain extent, its contribution to improving model performance gradually weakens. Through the selection process of the SPA algorithm, 228 feature variables were ultimately determined as the optimal feature subset.

#### 3.3.2. Feature Variable Extraction Based on UVE

Uninformative Variable Elimination (UVE) is a wavelength selection algorithm based on partial least squares regression coefficients. It identifies and eliminates variables that do not provide useful information by adding noise with disturbing spectral information to the model, thus reducing the number and complexity of variables in the model. The goal of this method is to simplify the input variables of the model and reduce its complexity while maintaining the predictive performance of the model.

To further screen the sample spectral wavelengths, we employed the UVE method. The training set and prediction set used by UVE are in the ratio of 3:1. The function called is plsuve(). Parameter settings: the optimal number of factors is 10%, the number of LOO times is 800, and the number of bands of random noise is 100. In practical applications, we placed the actual spectral variables on the left side of the vertical line in Figure 7, while the added random noise was placed on the right side. The horizontal dashed lines in Figure 7 set a threshold, and the variables located between these two dashed lines are considered uninformative variables that do not contribute to modeling the rice origin. Variables outside the dashed lines are selected as characteristic variables for modeling, and serve as inputs for subsequent analysis.

Through the screening process of the UVE method, as the number of runs increases, when the RMSE reaches its minimum, the number of selected features is 557, accounting for approximately 36.47% of the original data. This indicates that the UVE method effectively removes a large number of uninformative variables and retains key features closely related to the modeling objective.

## 4. Establishment of SVM Classification Model for Rice Origin

### 4.1. Recognition Results of the Model under Different Variable Selection Methods

For the filtered variables, we employed Support Vector Machine (SVM) as the classifier for modeling. To assess the stability and robustness of the model, we utilized a 10-fold cross-validation approach, excluding 25% of the training objects for testing in each iteration [18]. During this process, the samples for the validation and test sets were randomly partitioned.

Table 2 presents the model validation results under different methods. As can be seen from the table, the Continuous Projection Algorithm (SPA) achieved a low number of selected variables, a short running time, and high accuracy. This result indicates that SPA exhibits high efficiency in feature selection, able to significantly reduce the number of required variables while maintaining model performance. This contributes to simplifying the model structure, enhancing its interpretability, and mitigating the risk of overfitting.

By comparing the model validation results under different methods, we can conclude that the combination of SPA and SVM classifier demonstrates good performance when dealing with specific problems. This conclusion provides a useful reference for subsequent research and applications.

### 4.2. Model Effectiveness Validation

The classification results are presented in the confusion matrix. Figure 8 shows the confusion matrix for the SVM model’s predictions. Figure 8a displays the actual number of predicted samples for each seed category, while Figure 8b illustrates the overall accuracy across all samples. The recognition accuracy of the SVM model based on NZ-WT-SAP exceeds 99%, demonstrating strong generalization performance. This indicates that the NZ-WT-SPA-SVM model uses the minimum data dimension for modeling, achieves high classification accuracy, and has a short running time.

## 5. Conclusions

In this paper, fluorescence spectroscopy technology was used to identify the origin of different rice varieties that are invisible to the naked eye. After smoothing preprocessing of the data, the accuracy of the model decreased, which may be due to the loss of effective information during the smoothing process. Through normalization and denoising, the high-frequency noise and overlapping peak effect in the spectral data were significantly reduced. This paper used two feature extraction methods, SPA and UVE. Both methods can reduce data redundancy and improve model performance. In the experiment, the SVM model based on SPA showed a higher accuracy rate of 99.5%, which can effectively distinguish samples from eight rice-producing areas in Jilin Province (Gongzhuling, Hunchun, Jilin, Meihekou, Qianguo, Songyuan, Taoerhe, and Yushu). The experimental results show that, due to its fast and accurate characteristics, this method can provide researchers with faster and more effective tools for identifying the origin of rice, and is expected to be widely used in practical applications in the future.

## Figures and Tables

**Figure 1 sensors-24-02994-f001:**
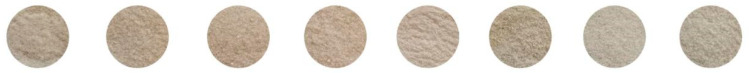
Diagram of samples to be tested. The order of rice producing areas from left to right is: Gongzhuling, Hunchun, Jilin, Meihekou, Qianguo, Songyuan, Taoerhe, and Yushu.

**Figure 2 sensors-24-02994-f002:**
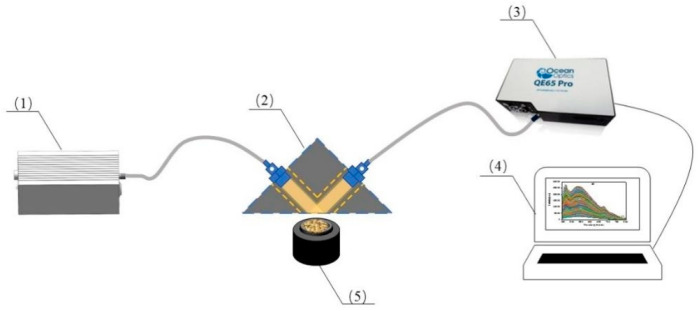
Fluorescence spectroscopy detection equipment diagram. (1): 405 nm laser light source; (2): 90° reflection head; (3): QE65pro spectrometer; (4): computer; (5): sample cuvette.

**Figure 3 sensors-24-02994-f003:**
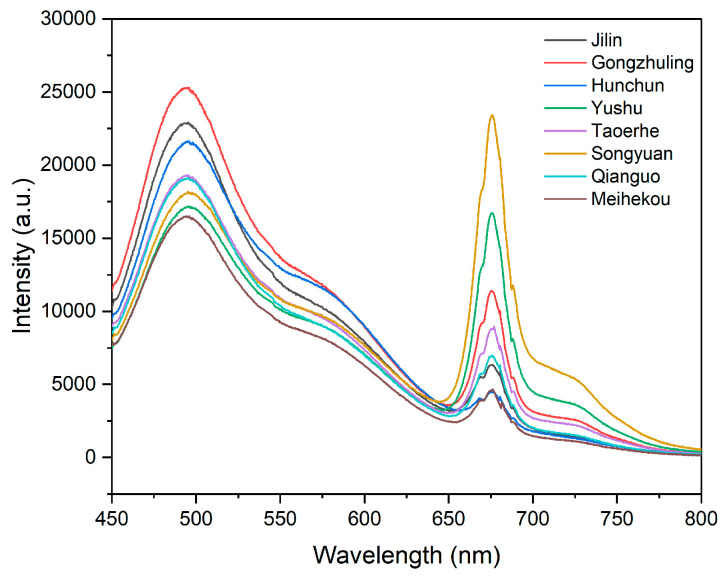
Original average reflectance fluorescence spectra of rice from different origins.

**Figure 4 sensors-24-02994-f004:**
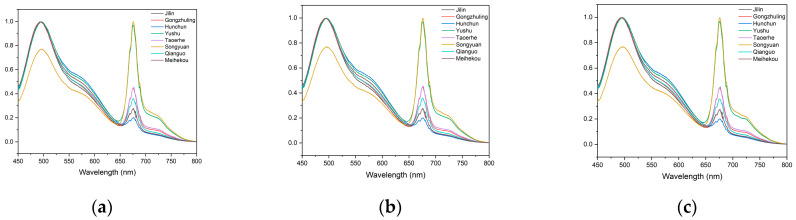
Preprocessing of fluorescence spectra of rice from different origins. (**a**) NZ, (**b**) NZ-SG, (**c**) NZ-WT.

**Figure 5 sensors-24-02994-f005:**
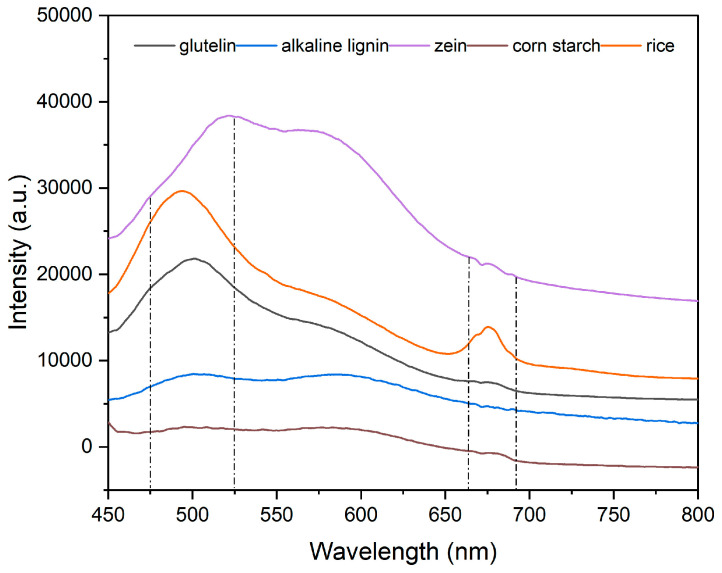
Spectral contrast diagram of samples with different components.

**Figure 6 sensors-24-02994-f006:**
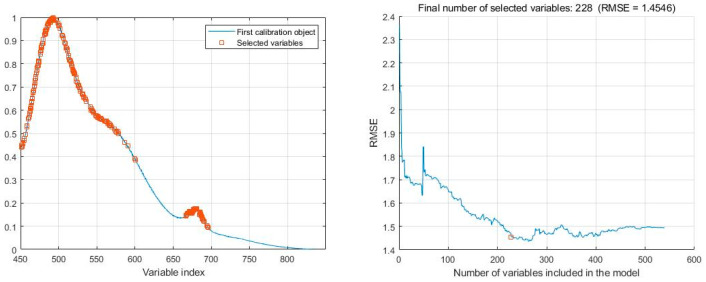
Process of selecting characteristic variables of SPA and distribution of selected wavelengths.

**Figure 7 sensors-24-02994-f007:**
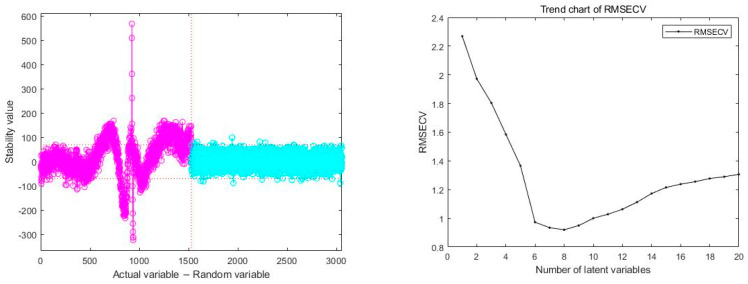
Trend of RMSE with number of runs.

**Figure 8 sensors-24-02994-f008:**
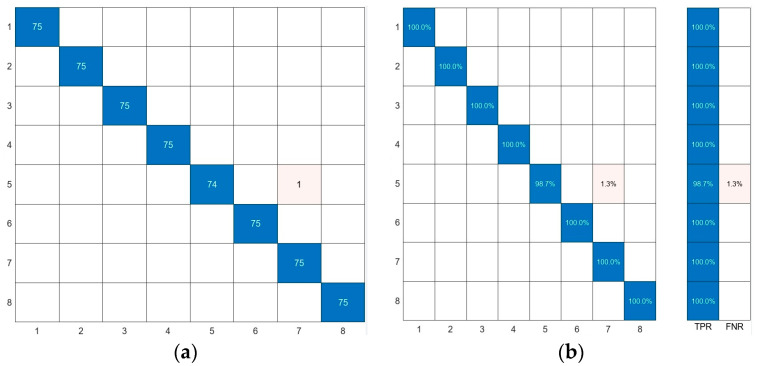
Confusion matrix of NZ-WT-SPA-SVM model prediction results. (**a**) Actual number of samples predicted for each type of seed, (**b**) overall accuracy.

**Table 1 sensors-24-02994-t001:** Modeling results of different pretreatment methods.

Serial Number	Pretreatment Method	Training Set			Test Set		
		R^2^	RMSE	MAE	R^2^	RMSE	MAE
1	NZ	0.9907	0.0060	0.0043	0.9577	0.0129	0.0106
2	NZ-SG	0.9885	0.0066	0.0050	0.9850	0.008	0.0061
3	NZ-WT	0.9959	0.0040	0.0033	0.9958	0.0039	0.0032

**Table 2 sensors-24-02994-t002:** SVM modeling results based on SPA and UVE features.

Discrimination Model	Feature Extraction Method	Data Dimension	Training Set Accuracy (%)	Test Set Accuracy (%)	Running Time (S)
SVM	Non	1527	100	99	9.1516
	SPA	228	99.83	99.5	2.1605
	UVE	557	99.83	99.5	3.1871

Note: “Non” indicates that no feature extraction method was used, which applies to the following cases as well.

## Data Availability

Due to the nature of this research, participants of this study did not agree for their data to be shared publicly, so supporting data is not available.

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
