# Peer review of "Rice Origin Tracing Technology Based on Fluorescence Spectroscopy and Stoichiometry"

_sensors, 2024, doi:10.3390/s24102994_

Round 1
Reviewer 1 Report
Comments and Suggestions for Authors
- The manuscript needs to be explained in a clear way,
- The cited references need to include more recent publications and relevant. There is no excessive number of self-citations. The references are not presented in a proper way through the text.
- The manuscript is presented scientifically sound. The experimental design is appropriate to test the hypothesis.
- The manuscript’s results are reproducible based on the details given in the methods section.
- The figures/tables/images, or schemes, are appropriate. They properly show the data, easy to interpret appropriately and consistently throughout the manuscript.
- The conclusions are consistent with the evidence and arguments.

The langauge needs further improvement.
Reviewer 2 Report
Comments and Suggestions for Authors
please kindly revise followed my comment in attachment

Author Response
Thank you for your advice!
Please see the attachment.

Reviewer 3 Report
Comments and Suggestions for Authors
Author Response

(The authors gave the same response as above.)

Reviewer 4 Report
Comments and Suggestions for Authors
The paper primarily presents the test results without delving into analysis and discussion. Nonetheless, its lack of innovation is evident, given the extensive exploration of the topic in previous studies. Therefore, it is imperative to conduct and elucidate upon the changes proposed within this study.
Detailed comments are listed below:
1. The sentences in the summary are too long or complex. It is recommended that the sentence structure be simplified to improve readability and comprehension.
2. Introduction has introduced other rapid detection methods, what are the advantages of other rapid detection methods compared with fluorescence detection? What is the progress of fluorescence spectroscopy?
3. Line 118-120. Are the references cited in the article formatted correctly?
4. The preprocessed spectrogram confuses me. b and c in Figure 4 are the same? And why is the preprocessed spectrum more difficult to distinguish with the naked eye than the original spectrogram?
Comments on the Quality of English LanguageMinor editing of English language required.
Author Response
Thank you very much for your valuable suggestions on this article.
- The sentences in the abstract have been revised.
- The introduction has added references related to fluorescence spectroscopy.
- The format of the references is incorrect and has been corrected.
- Due to the limited space of the document, the images are small. The spectral curves of the preprocessed images are smoother and less burry compared to the original images, and the accuracy of model evaluation has improved. b and c in Figure 4 are not the same. b is smoothing preprocessing, and c is wavelet transform processing.
These issues are all reflected in the attachment.

Round 2
Reviewer 4 Report
Comments and Suggestions for Authors
The authors have made systematic revisions to the manuscript in accordance with comments of the reviewers. I think it can be accepted in its current form.